# Stress-induced plasticity of a CRH/GABA projection disrupts reward behaviors in mice

Matthew T. Birnie [1,2,5], Annabel K. Short [1,2,5], Gregory B. de Carvalho [2], Lara Taniguchi [1,2], Benjamin G. Gunn [2], Aidan L. Pham[1,2], Christy A. Itoga [2], Xiangmin Xu[2], Lulu Y. Chen [2], Stephen V. Mahler [3], Yuncai Chen [1,2] ✉ & Tallie Z. Baram [1,2,4] ✉

Disrupted operations of the reward circuit underlie major emotional disorders, including depression, which commonly arise following early life stress / adversity (ELA). However, how ELA enduringly impacts reward circuit functions remains unclear. We characterize a stress-sensitive projection connecting basolateral amygdala (BLA) and nucleus accumbens (NAc) that co-expresses GABA and the stress-reactive neuropeptide corticotropin-releasing hormone (CRH). We identify a crucial role for this projection in executing disrupted reward behaviors provoked by ELA: chemogenetic and optogenetic stimulation of the projection in control male mice suppresses several reward behaviors, recapitulating deficits resulting from ELA and demonstrating the pathway's contributions to normal reward behaviors. In adult ELA mice, inhibiting–but not stimulating–the projection, restores typical reward behaviors yet has little effect in controls, indicating ELA-induced maladaptive plasticity of this reward-circuit component. Thus, we discover a stress-sensitive, reward inhibiting BLA → NAc projection with unique molecular features, which may provide intervention targets for disabling mental illnesses.

The brain is organized in circuits that orchestrate complex behaviors, including pleasure and reward[1]. The nucleus accumbens (NAc) is a major node of the reward circuit and is involved in pleasure and motivation processes[2]. Multiple afferents converge on the NAc to modulate reward behavior, including from the basolateral amygdala (BLA)[3,4]. The BLA mediates associative learning of aversive and appetitive stimuli, and glutamatergic BLA projections to the NAc facilitates reward behavior[5–7].

The maturation of brain circuits during the early postnatal period is influenced by experiences including stress[8–11]. Stress experienced during critical early-life periods (ELA) can exert enduring effects on brain circuits and their functions, resulting in mental health problems[12–14]. The mammalian brain is endowed with evolutionarily conserved stress-sensitive molecules, expressed in specific regions/brain-circuit nodes. Therefore, we reasoned that the

effects of early-life stress on reward circuit operation might take place by influencing the expression or function of neurons and projections expressing such stress-sensitive molecules. We focused on the neuropeptide CRH, an orchestrator of neuroendocrine stress signaling[15] and an established modulator of NAc functions in a stress-dependent manner[16,17]. In addition, CRH-expression and function are enduringly influenced by early-life adversity (ELA) in several brain regions[18–20]. Therefore, considering the peptide a potential marker or an effector in stress-sensitive neurons, we mapped CRH-expressing afferent projections to the NAc. Here we describe a GABAergic CRH+ projection from the BLA → NAc and the use of chemogenetic and optogenetic strategies to determine its role in regulating reward behavior. We then establish that this projection mediates deficits in adult reward-seeking behaviors that are observed following ELA.

[1]Department of Pediatrics, University of California-Irvine, Irvine, CA, USA. [2]Department of Anatomy/Neurobiology, University of California-Irvine, Irvine, CA, USA. [3]Department of Neurobiology & Behavior, University of California-Irvine, Irvine, CA, USA. [4]Department of Neurology, University of California-Irvine, Irvine, CA, USA. [5]These authors contributed equally: Matthew T. Birnie, Annabel K. Short. ✉e-mail: yuncaic@uci.edu; tallie@uci.edu

## Results

### Characterization of a CRH/GABA projection from the BLA → NAc

First, in validated (Supp. Fig. 1) CRH-ires-CRE mice[21] we injected a CRE-dependent retrograde virus into the medial NAc shell (Fig. 1a, b) to retrogradely infect projection neurons[22,23], including those originating in the BLA (Fig. 1c). The specificity of CRE-dependent viral expression to CRH[+] neurons was confirmed with colocalization of endogenous CRH in viral-infected BLA cell bodies (Fig. 1d). To define the neuroanatomical distribution of the BLA-origin CRH[+] projection in the NAc, we injected a CRE-dependent anterograde virus into the medial BLA (Fig. 1e–g,) and identified CRH[+] BLA projection fibers and terminals in the medial NAc shell (Fig. 1h). A combination of viral reporter, fluorescent in situ hybridization and immunostaining

identified this population of CRH[+] neurons as GABAergic, rather than glutamatergic (Fig. 1i–k). Specifically, shown are images of BLA neurons from CRH ires-CRE mice injected with a CRE-dependent retrograde virus injected into the medial NAc shell to express tdTomato in BLA-origin cell bodies (Fig. 1i). These CRH[+] neurons co-express two defining GABA markers: Gad67 (Fig. 1i) and the vesicular GABA transporter-vGAT (Fig. 1j). In contrast, they do not express a defining glutamatergic marker-CaMKII (Fig. 1k).

### CRH[+] BLA → NAc projection stimulation evokes GABA currents only

To substantiate that the CRH[+] BLA → NAc projection evokes inhibitory currents, we performed whole-cell patch-clamp recordings in the

**Fig. 1 | A projection of CRH/GABA neurons in the medial BLA to the medial NAc shell. a–c** Retrograde tracing of CRH[+] neuronal inputs to medial NAc shell of CRH-ires-CRE mice. **a** Schematic of construct and injection location of AAV2-retro-CAG-FLEX-tdTomato-WPRE virus that permits retrograde access to projection neurons providing afferent inputs to NAc. **b** Example confocal micrograph of locally infected CRH[+] axon terminals in medial NAc shell. **c** Retrograde tracing identifies the medial BLA as a robust source of CRH[+] NAc inputs. **d** 3D image (z-stack; 0.5 μm steps) confirmed localization in the BLA of AAV-retro infected cells (red) that co-express endogenous CRH (green); dual labeled neurons = yellow. **e–g** Anterograde tracing of CRH[+] axonal projections from BLA to medial NAc shell. **e**, The AAV1-DIO-tdTomato construct and the viral genetic experimental design. **f** Virus injection is confined to the BLA, **g** and absent from the central amygdala (CeA), shown by selective expression of tdTomato in BLA CRH[+] neurons. **h** BLA-origin CRH[+] axons and terminals in the medial NAc shell. **i–k** Virus injection into the medial NAc shell retrogradely infected somata in the BLA. **i** Combined fluorescence in situ hybridization (FISH) and immunostaining with GAD67 mRNA in CRH[+] cells in the BLA. Arrowheads point to co-localized GAD67 mRNA and virus-reporter labeling. **j** a BLA → NAc cell (red) co-expresses endogenous CRH (green) and vGAT (magenta), but **k** does not co-express the glutamatergic marker CaMKII. ** = Major Island of Calleja, ac anterior commissure, DB diagonal band. Scale bars in **i** and **k** = 10 μm. To confirm findings, virus injections, projection assessment, and immunohistochemistry were assessed in mice from at least two independent litters.

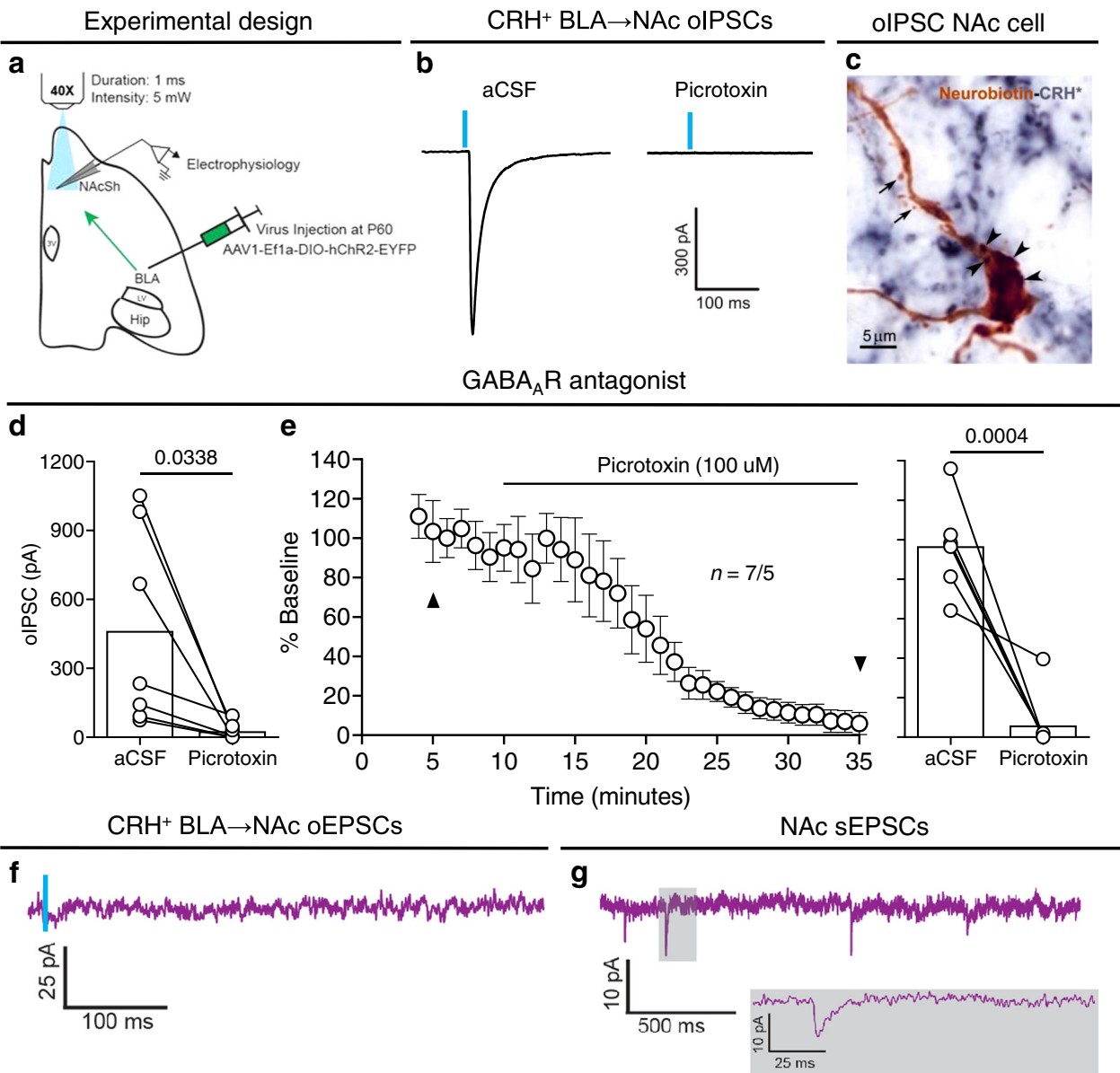

**Fig. 2 | Optical stimulation of CRH⁺ BLA-origin axons in the NAc evokes exclusively IPSCs. a** Schematic of experimental design for electrophysiology recordings in the whole-cell patch-clamp configuration. Horizontal brain slices containing BLA and NAc from CRH-ires-CRE mice that were injected with Cre-dependent ChR2-EYFP in BLA. **b** Representative traces of optically evoked IPSCs (oIPSCs). These were blocked in the presence of GABA_A receptor antagonist picrotoxin. **c** A neurobiotin (brown) filled neuron from which oIPSCs were recorded. Note spines (arrows) suggesting the cell is a medium spiny neuron (MSN). Arrowheads denote ChR2-expressing boutons from BLA-origin CRH⁺ axons (CRH; blue) on the soma. **d** oIPSCs amplitudes pre and post picrotoxin ($n = 7$ neurons, 5

mice). **e** Time-course plot of normalized oIPSCs amplitudes throughout the recording and following application of picrotoxin. Black triangles in **e** denote trace recordings in **b** and timepoint analysis in **d** and **e**. **f** Representative trace of a NAc cell showing no response from optically evoked EPSC (oEPSC) at −70 mV, obtained after verifying oIPSCs at 0 mV in the presence of picrotoxin. **g** Representative trace showing spontaneous EPSCs (sEPSCs) were still present; gray box shows magnified view recording. In **d** and **e**, bars represent mean. In **e** circles represent mean ± SEM. Two-sided paired *t*-tests in **d** and **e**. **d** oIPSCs (current) pre vs. post PTX: $P = 0.0338$. **e** oIPSCs (normalized) pre vs. post PTX: $P = 0.0004$. 3 V = third ventricle, Hip Hippocampus, LV lateral ventricle. Source data are provided as a Source Data file.

medial NAc shell of CRH-ires-CRE mice injected with a CRE-dependent ChR2-EYFP in the BLA (Fig. 2a). Optical stimulation of CRH⁺ BLA-origin fibers in the NAc reliably evoked inhibitory postsynaptic currents (oIPSCs) (Fig. 2b, d) in medium spiny neurons (MSNs) (Fig. 2c). The optically evoked inhibitory postsynaptic responses were consistently blocked by superfusion with the GABA_A receptor antagonist, picrotoxin (100 μM) (Fig. 2b, d, e). In addition, in a confirmed oIPSC recorded cell, optical stimulation did not yield excitatory postsynaptic currents (oEPSCs) (Fig. 2f), even though spontaneous EPSCs were apparent (Fig. 2g). Together, these data identify an inhibitory CRH/GABA BLA→NAc projection that is distinct from the well-described glutamatergic BLA→NAc projection[5,6,24].

## CRH⁺ BLA → NAc projection activity suppresses reward behavior

To investigate the function of this CRH/GABA BLA → NAc projection, we targeted it during reward behaviors using chemogenetic and optogenetic strategies. Specifically, in typically reared male and female mice, we microinjected CNO[25,26] into the medial NAc shell (Fig. 3d, e; Suppl. Fig. 4). We stimulated the hM3Dq⁺ CRH/GABA BLA-origin fibers in the NAc during two different reward tasks—one centered on palatable food consumption and the other involving sex cues[27,28]. Bilateral microinfusion of CNO (1 mM) into the medial NAc shell reduced palatable food consumption in male but not female mice (Fig. 3d). This suppression, unique to males was also evident in the sex-cue task (Fig. 3e). Inhibition of the projection in TR male mice had no effect, and

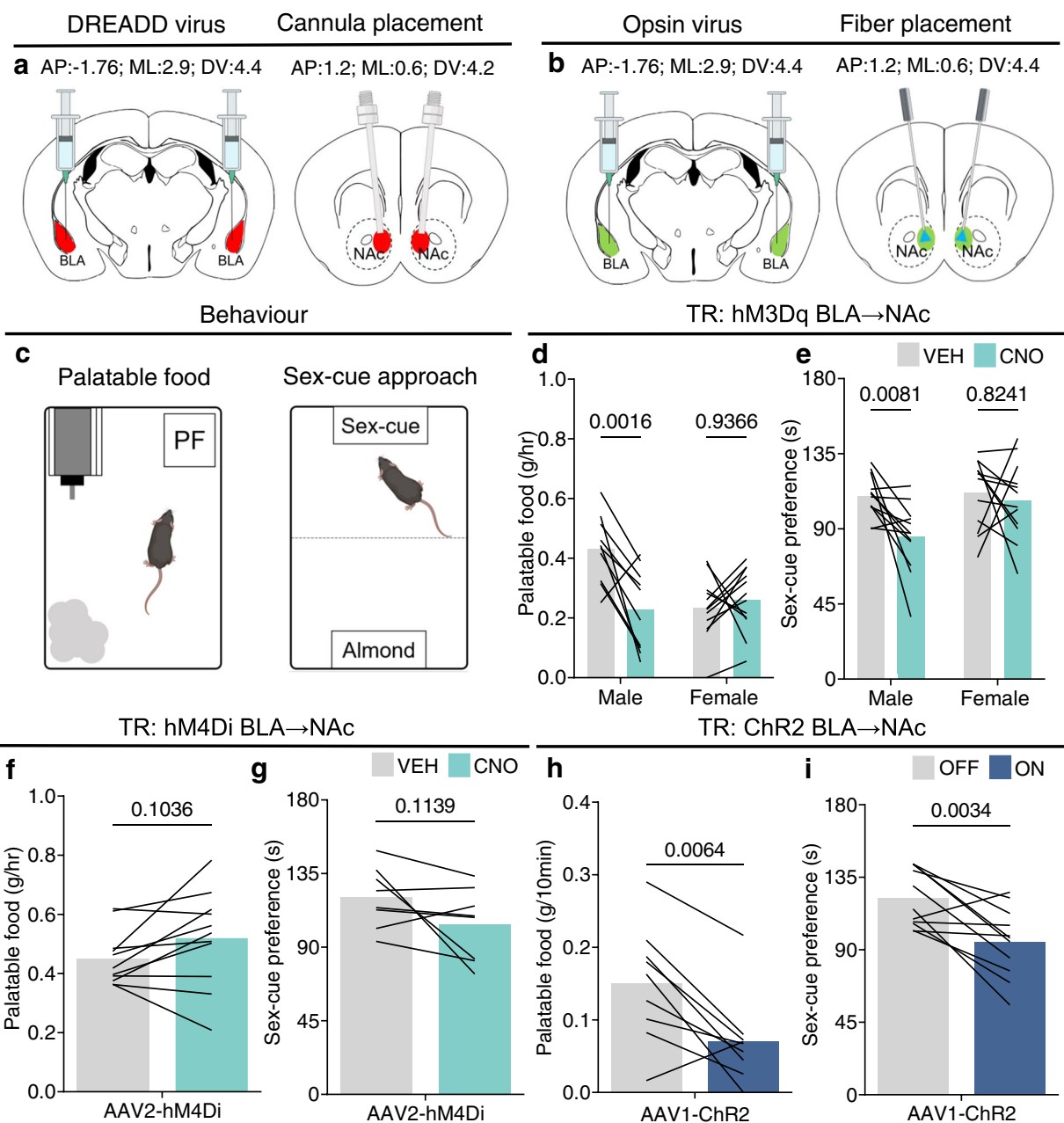

**Fig. 3 | Stimulating the CRH/GABA BLA → NAc projection suppresses reward in typically reared male mice. a**, **b** Coordinate locations of **a**, DREADD injection and CNO infusion and **b** opsin injection and fiber placement. **c** schematic of the reward tasks. **d**, **e** Chemogenetically stimulating the CRH/GABA BLA → NAc projection with microinfusion of CNO in the medial NAc shell suppressed **d**, palatable food consumption ($n = 10$ male mice; 12 female mice) and **e** preference for a sex-cue ($n = 12$ male mice; 12 female mice) in males, but not females. **f**, **g** Inhibiting the CRH/GABA BLA → NAc projection in TR male mice did not increase **f**, palatable food consumption ($n = 11$ mice) or **g**, preference for a sex-cue ($n = 8$ mice). **h**, **i** stimulating the CRH/GABA ChR2-expressing BLA → NAc projection decreased **h** palatable food

consumption ($n = 9$ mice) and **i**, approach time to a sex-cue ($n = 11$ mice). In **d**–**i** bars represent mean. Two-way ANOVA with repeated measures followed by post hoc tests (**d**, **e**). **d** hM3Dq BLA → NAc: Sex x Treatment−$F = 5.691$, DFn = 1, DFd = 40, $P = 0.0219$; post hoc with Tukey's multiple comparison (Veh vs. CNO: Male−$P = 0.0016$; Female−$P = 0.9366$). **e** hM3Dq BLA → NAc: Treatment−$F = 6.448$, DFn = 1, DFd = 44, $P = 0.0147$; post hoc with Sidak's multiple comparison (Veh vs. CNO: Male−$P = 0.0081$; Female−$P = 0.8241$). Two-sided paired $t$-tests (**f**–**i**). **f** hM4Di BLA → NAc: $P = 0.1036$; **g** hM4Di BLA → NAc: $P = 0.1139$. **h** ChR2 BLA → NAc: $P = 0.0064$; **i** ChR2 BLA → NAc: $P = 0.0034$. PF palatable food. Gray = vehicle/light off, teal = CNO, blue = light on. Source data are provided as a Source Data file.

specifically did not increase palatable food consumption (Fig. 3f) or the preference for a sex cue (Fig. 3g). Notably, these outcomes were not the result of off-target effects of CNO (Suppl. Fig. 2d), as non-reward behaviors were unaffected. We confirmed the activation of hM3Dq-expressing CRH⁺ neurons by CNO via quantification of selective Fos expression in the BLA 90 min following injection (Suppl. Fig. 2c). We then used a second, independent technology, optogenetics, injecting a CRE-dependent ChR2-EYFP in the BLA bilaterally to

CRH-ires-CRE mice, and implanting bilateral optic fibers directly above the medial NAc shell to optically stimulate projection fibers (Fig. 3b). Blue light (473 nm) stimulation (10 Hz) suppressed palatable food consumption (Fig. 3h) and the preference for a sex-cue in males (Fig. 3i). Again, potential off-target effects of virus and/or light stimulation did not contribute to these outcomes (Suppl. Fig. 2e−h). Collectively, these findings describe an unexpected reward-suppressing role for a CRH/GABA BLA → NAc projection in males, which differs from

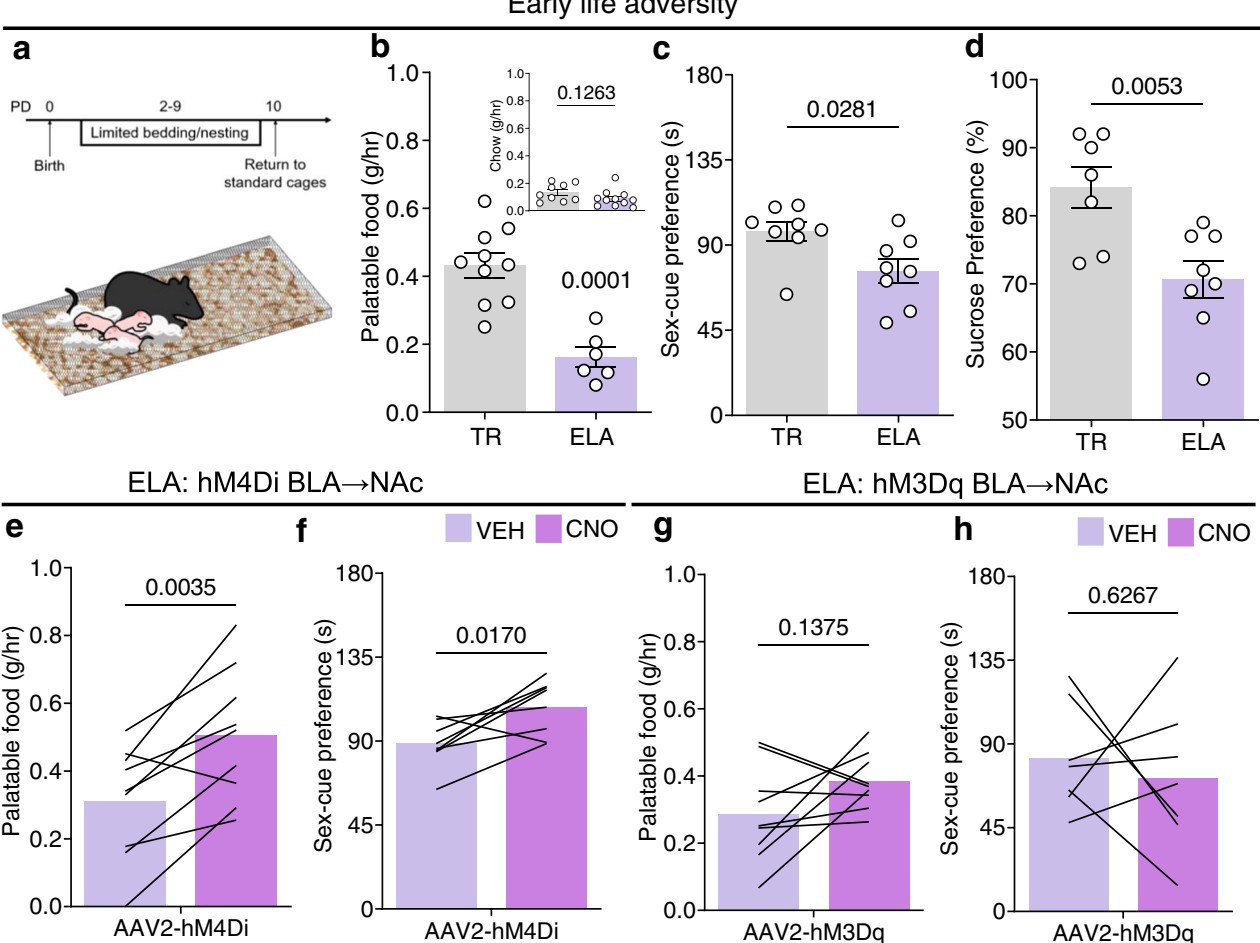

**Fig. 4 | Inhibiting the CRH/GABA BLA → NAc projection rescues reward deficits following early-life adversity. a** Timeline and environment (limited bedding and nesting) of ELA. **b–d** ELA reduced palatable food consumption (*n* = 16 mice; *TR* = 10, *ELA* = 6), yet did not alter regular chow consumption (*n* = 20 mice; *TR* = 9, *ELA* = 11). ELA reduced preference for a sex-cue (*n* = 16 mice; *TR* = 8, *ELA* = 8) and preference for sucrose (*n* = 15 mice; *TR* = 7, *ELA* = 8). **e** Inhibiting the hM4Di⁺ CRH/GABA BLA → NAc projection with microinfusion of CNO in the medial NAc shell increased palatable food consumption (*n* = 9 mice) and **f** preference for a sex-cue (*n* = 8 mice). However, stimulating the hM3Dq⁺ CRH/GABA BLA → NAc did not influence

**g** palatable food consumption (*n* = 9 mice) or **h** the preference for a sex-cue (*n* = 7 mice). In **b–d**, bars represent mean ± SEM; in **e–h** bars represent mean. Two-sided unpaired *t*-tests (**b**, **d**), two-sided unpaired *U*-test (**c**), two-sided paired *t*-tests (**e–h**). **b** TR vs. ELA main: *P* = 0.0001, inset: *P* = 0.1263; **c** TR vs. ELA: *P* = 0.0281; **d** TR vs. ELA: *P* = 0.0053; **e** ELA hM4Di BLA → NAc: *P* = 0.0035; **f** ELA hM4Di BLA → NAc: *P* = 0.0170; **g** ELA hM3Dq BLA → NAc: *P* = 0.1375; **h** ELA hM3Dq BLA → NAc: *P* = 0.6267. Gray = typical reared, mauve = ELA/ELA + vehicle, pink = ELA + CNO. Source data are provided as a Source Data file.

the well-established, reward enhancing glutamatergic projection connecting the BLA and NAc[5,6]. As CRH expression and function are often modulated by stress[16,19,29], these data support the possibility that this projection, acting via GABA, CRH, or both neuromodulators, might mediate suppressed reward behaviors often observed after adult or developmental stresses.

### Inhibiting CRH⁺ BLA → NAc activity rescues reward behavior after ELA

A large body of work documents a disruption of reward behaviors following ELA in both humans and experimental models[30–34]. To determine whether the CRH/GABA BLA-NAc projection contributes to these deleterious effects, we first determined consequences of ELA on reward behaviors in males. We used a well-established model of ELA during a sensitive developmental period[35,36] (Fig. 4a), in which poverty is simulated via a resource-scarce environment during postnatal days 2–10 in mice[12]. In adult ELA mice, preference for palatable food, but not regular chow, was diminished (Fig. 4b). Similarly, preference for a sex-cue (Fig. 4c) and sucrose (Fig. 4d) were also reduced, whereas effect of ELA on the open field or swim stress tasks were not apparent

(Suppl. Fig. 3a, b). Importantly, the reward deficits in adult ELA mice recapitulated the effects of chemogenetic and optogenetic stimulation of the CRH/GABA BLA → NAc projection in typically reared male mice (Fig. 3), identifying a possible role for this projection in ELA-induced reward deficits[9]. To directly test this, we inhibited the activity of the CRH/GABA BLA → NAc projection during reward tasks in adult ELA mice. Specifically, we provided CRH-ires-CRE ELA mice with bilateral injections of a CRE-dependent inhibitory DREADD (hM4Di) into the BLA and then infused CNO (1 mM) directly into the medial NAc shell (Fig. 4e, f). Inhibition of the BLA-origin fibers in the NAc rescued the consumption of palatable food in adult ELA-experienced mice (Fig. 4e) and restored the preference for a sex-cue (Fig. 4f). We then assessed if stimulation of this projection further suppressed reward behavior in ELA mice by microinfusing CNO directly into the medial NAc shell in CRH-ires-CRE mice expressing hM3Dq in the CRH⁺ BLA-NAc projection. CNO did not further decrease palatable food consumption (Fig. 4g) or double the preference for a sex cue (Fig. 4h). Non-reward behaviors were not influenced by any manipulation (Suppl. Fig. 3c, d). Together, these data suggest that inhibiting the CRH/GABA BLA → NAc projection in ELA, but not TR mice, increases

reward behavior indicating a selective ELA-induced maladaptive plasticity of this reward-circuit pathway.

## Discussion

Exposure to severe or chronic stressors during sensitive periods in development can produce profound and often lasting changes to brain operations that can be detrimental to mental health[13,37]. A majority of the world's adults have experienced a form of ELA, which precedes–and likely contributes to–affective disorders including depression, substance use and obsessive risk-taking[12,38–41]. These major public health challenges are associated with aberrant reward processing[42,43]. Remarkably, the cellular and circuit foundations of ELA-associated disruption of reward-related behaviors have remained largely elusive[12]. In this study, we first discovered an inhibitory BLA → NAc projection responsible for ELA-provoked deficits in reward behavior. Unlike the canonical glutamatergic projection from BLA → NAc[5,6], this stress-sensitive CRH/GABA BLA → NAc projection suppresses several types of reward behaviors in typically reared male, but not female, mice. Whereas future studies are needed to identify GABA, CRH or interaction of both neuromodulators as key effectors in this projection, our findings highlight the emerging complexity of reward circuit operations including cell-type-specific projections that anatomically parallel established connections (e.g., BLA to NAc), yet exert different and sometimes opposing (and sex-specific) effects on behavior.

This work also identifies the BLA as a locus of neurons expressing CRH, a peptide known to be released during stress in several brain regions[16,17,29,44]. Indeed, most studies on amygdalar CRH have centered on the central nucleus[45–48], which is rich in CRH-expressing cells yet does not project directly to the NAc[23] (Fig. 1). Notably, while involved in mediating the effects of stress in some brain regions, local infusion of CRH in the NAc, instead increases motivation for sucrose[49], and severe stress switches the action of CRH in the NAc from appetitive to aversive[16,50]. Taken together, these studies reiterate the complexity of intercalating stress and reward systems and the important need to consider cell type(s), the origin and termination of projections, their multiple neurotransmitters/peptides, and the sex and biological context of the electrophysiological and behavioral measures under study.

To conclude, we have delineated a substantial population of CRH/GABA neurons in the BLA which project monosynaptically to the NAc to influence reward behavior, and importantly, whose function is impacted by ELA, promoting pathological deficits in these behaviors. Advancing our knowledge of stress-mediated control of reward behavior takes us a step closer to understanding the mechanisms of several prevalent major mental illnesses and provide new targets for the prevention and intervention of such stress-related disorders.

## Methods

### Experimental animals

Male and female B6(Cg)-Crh[tm1(cre)Zjh]/J (CRH-ires-CRE) mice were used in experimental procedures. Food and water were available ad libitum. Mice were bred in house from Jackson Laboratories stock (Stock no: 012704). All mice were housed in standard conditions at 72 °F and 42% humidity and on a 12-h light–dark cycle (Lights ON: 0600; Lights OFF: 1800). All mice were single housed following surgical procedures for experimental purposes. All experimental procedures were approved by the University of California-Irvine Institutional Animal Care and Use Committee (AUP 21-128 and 18-183) and were in accordance with the guidelines from the National Institute of Health.

### Limited bedding and nesting paradigm

Early-life adversity was induced on postnatal day (PN) 2 to 9 using our laboratory's standard protocol[35,36]. Briefly, routine plastic mouse cages (L37 x W20 x H17 cm) were fitted with an aluminum mesh (cat # 4700313244, McNichols, US) platform sitting ~2.5 cm

above the cage floor. Bedding was reduced to cover the cage floor sparsely, and one-half of a single nestlet (5 × 2.5 cm) was provided for nesting material. Typically reared (TR) dams and pups resided in standard bedding cages, but were moved to new cages on P2 to control for handling during the LBN setup. TR and LBN cages were undisturbed from P2-P9, and housed in a quiet room. On P10, both TR and ELA (LBN) groups were transferred to fresh, routine cages.

### Animal surgery

Mice (PN 60) were anaesthetized with either ketamine/xylazine (100/10 mg/kg body weight) or isoflurane (1–1.5%) for stereotaxic surgery. For retrograde tracing experiments, mice received a unilateral injection (0.2 μl, 20–30 nl/min) of AAV2-retro-CAG-FLEX-tdTomato-WPRE into the left medial NAc shell (AP: 1.18, ML: −0.6, DV: 4.4). For anterograde tracing experiments, mice received a unilateral injection (0.2 μl) of AAV1-DIO-tdTomato into left hemisphere BLA (AP: −1.76, ML: −4.19 at 15° oblique, DV: 3.92). For in vivo chemogenetic excitation during behavior testing, mice were injected with the AAV2-hM3D(Gq)-mCherry virus or the same viral vectors carrying mCherry alone (5 × 10^{12} vector genomes/mL) bilaterally into the BLA (AP: −1.76, ML: ± 4.19 at 15° oblique, DV: 3.92). For in vivo chemogenetic inhibition during behavior testing, mice were injected with the AAV2-hM4D(Gi)-mCherry bilaterally into the BLA (AP: −1.76, ML: ± 4.19 at 15° oblique, DV: 3.92). For intra NAc drug infusion, guide cannulae were implanted over the medial NAc shell (AP: 1.18, ML: ± 0.6, DV: 4.2). For in vivo optogenetic excitation during behavior, and in vitro electrophysiology experiments, mice were injected with the AAV1-EF1a-DIO-hChR2(h134R)-EYFP virus or the same viral vector carrying EYFP alone, bilaterally into the BLA. For projection specific targeting during behavior testing, optic fibers (0.22 NA, 100 μm diameter) were placed dorsal to EYFP-expressing axonal terminals in the medial NAc shell (AP:1.18, ML: ± 0.6, DV:4.4).

### Chemogenetic studies

For excitatory (AAV2-hM3D(Gq)-mCherry) and inhibitory (AAV2-hM4D(Gi)-mCherry) DREADD experiments: Intraperitoneal injection of 1 mg/kg Clozapine-N-Oxide (CNO) (NIMH, # MH-929, Batch #14073-1), CNO was dissolved in 1% Dimethyl sulfoxide (DMSO) in phosphate buffer saline (PBS). For microinfusion of CNO (CNO dihydrochloride, cat #HB6149, Batch #E1169-1-2, HelloBio, UK) into the medial NAc shell, 1 mM CNO (0.2 μl/side) in artificial cerebral spinal fluid (in mM: 124 NaCl, 3 KCl, 26 NaHCO_3, 1.4 NaH_2PO_4, 1 MgSO^4, 10 D-Glucose, 2 CaCl_2).

### Optogenetic studies

For in vivo experiments, <5 mW of blue light (95–159 mW/mm^2 at the tip) was generated by 473-nm μLED (Prizmatix, US), and bilaterally delivered through two fiber optic patch cords (0.22 NA, 200 μm diameter, Prizmatix, US). Light delivery was controlled using a pulse generator (PulserPlus, Prizmatix, US) to deliver 5 ms light pulse trains at 10 Hz[6] during behavior testing.

For ex vivo electrophysiology recordings, <5 mW of blue light was generated by a 473-nm LED (Sola Light Engine, Lumencor, US) and delivered through a 40× objective. Frequency (0.1 Hz) and pulse duration (1 ms) were controlled by LLE Sola-SE2 software (Lumencor, US).

### Electrophysiology

**Slice preparation.** CRH-ires-Cre mice were injected with AAV1-Ef1a-DIO-hChR2-EYFP in the basolateral amygdala (BLA) were deeply anesthetized with isoflurane and quickly decapitated (PN 120). Acute horizontal slices (300 μm) containing both basolateral amygdala and nucleus accumbens were obtained using a vibratome (V1200S, Leica, U) in ice-cold N-Methyl D-Glucamine (NMDG) cutting solution: containing (in mM): 110 NMDG, 20 HEPES, 25 glucose, 30 NaHCO_3 1.2

$NaH_2PO_4$, 2.5 KCl, 5 sodium ascorbate, 3 sodium pyruvate, 2 Thiourea, 10 $MgSO_4-7\ H_2O$, 0.5 $CaCl_2$, 305-310 mOsm, pH 7.4. Slices equilibrated in a homemade chamber for 25–30 min (31 °C) and an additional 45 min in room temperature aCSF containing (in mM): 119 NaCl, 26 $NaHCO_3$, 1 $NaH_2PO_4$, 2.5 KCl, 11 Glucose, 10 Sucrose, 2.5 $MgSO_4-7\ H_2O$, and 2.5 $CaCl_2$ (pH 7.4) before being transferred to a recording chamber. Recording solution was the same as equilibration solution only with 1.3 mM $MgSO_4-7\ H_2O$. All solutions were continuously bubbled with 95% $O_2$/5%$CO_2$.

**Whole-cell patch-clamp.** Whole-cell patch-clamp recordings were obtained in the medial shell of the NAc (NAcSh). Data were collected with a Multiclamp 700B amplifier, Digidata 1550B (Molecular Devices, US), and using Clampex 11 (pClamp; Molecular Devices, US). All recordings were acquired in voltage clamp at 34° Celsius and were digitized at 10 kHz and low pass filtered at 2 kHz. Patch pipette was filled with internal solution containing (in mM): 143 CsCl, 10 HEPES, 0.25 EGTA, 5 Phosphocreatine, 4 MgATP, 0.3 NaGTP (295–305 mOsm, pH 7.4 with CsOH) and 1 mg/ml Neurobiotin (cat # SP-1120, Vector Labs, US). CNQX (50 μM, cat # 0190, Tocris Biosciences, UK) and AP5 (10 μM, cat # 0106, Tocris Biosciences, UK) were added to aCSF to block glutamate receptors. All pipettes (3–4 MΩ) were pulled from borosilicate glass (cat # PC-100, Narishige, US). Series resistance (Rs) was monitored throughout the recording for patch sealing. Once whole-cell configuration was obtained, holding potential was set at −70 mV and a 1 ms, 5 mW light pulse was delivered every 10 s (0.1 Hz) through a 40× objective. Once current response was obtained, picrotoxin (100 μM, cat # P1675, Sigma-Aldrich, US) was washed in the recording chamber to verify that optically evoked inhibitory postsynaptic current (oIPSC) was mediated by GABAa receptor activation. Once recording was complete slices were transferred to 4% PFA overnight at 4° Celsius then to 0.1 M PB for post hoc processing of Neurobiotin (cat # SP-1120, Vector Labs, US)[51].

### Behavioral assays

All training and testing took place in standard housing mouse cages (34 × 18 cm) (home cage), apart from open field (45 × 45 cm) and swim stress. All behavior was carried out in low light (<15 lux) and during the animal's active period. Prior to testing start, all mice were placed in the behavior room for at least 1 h to acclimatize to their surroundings.

**Sucrose preference task.** Single-housed mice were habituated to the presence of two drinking bottles (both containing water) in their home cage for 3 days. Following acclimatization, one of the bottles was replaced with 1% sucrose solution and intake was measured 4 h after IP CNO injection for 3 consecutive days during the early active period. The bottles switched sides daily to prevent side bias. Total consumption of water and sucrose solution was measured at the end of each time session by weighing the bottles. Sucrose preference was defined as the ratio of the consumption of sucrose solution vs. the consumption of both water and sucrose solution combined. Testing was performed over 3 consecutive days. For chemogenetic experiments, mice were injected 1 h prior to the start of the 4 h test session.

**Sex-Cue approach.** For studies in male mice, urine was collected from females in the estrus phase of the estrous cycle. For assessing interest and preference of sex cues in female mice, male urine was collected. For both sexes, urine was collected on the day of the test and stored in 0.2 ml PCR tubes and capped until used. 60ul of urine and 60ul of a non-reward scent (Almond; Pure Almond extract, Target) were pipetted onto cotton tip applicators and fixed to the home cage of single-housed mice at either end. The mouse was given free access to both tips for 3 min. For chemogenetic experiments, IP CNO or intra NAc CNO was administered to the mice prior to test start (>1 h and <5 min,

respectively). For optogenetic experiments, mice were habituated to cable attachment prior to testing (>15 min). 3-point animal tracking was used to measure preference for scent with Ethovision XT15 (Noldus, US) software.

**Palatable food consumption.** Single-housed mice were habituated to ~1 g of Cocoa Pebbles cereal (Post, USA) in their home cage for 3 days prior to testing, also conducted in their home cage. Pre-weighed Cocoa Pebbles (~1 g) were placed in their cage, and intake was measured following 1 h (chemogenetic manipulation) or 10 min (optogenetic manipulation). For chemogenetic experiments, IP CNO or intra NAc CNO was administered to the mice prior to start (>1 h and <5 min, respectively). For optogenetic experiments, the mice were habituated to cable attachment prior to test start (>15 min).

**Open field.** During the active phase, and in low light (<15 lux measured on arena floor), mice were placed in one corner of an open field arena (L43 × W43 × H35 cm) and given free access to explore for 10 min. Animal movement was detected with an automated 3-point software (Ethovision XT15, Noldus, US). For analysis, the arena was subdivided into sixteen equal zones (4 inner and 12 outer squares). Ethovision XT15 (Noldus, US) software calculated the time spent in each of the 16 zones. To measure time spent in center, the 4 inner zones were collapsed together.

**Swim stress.** Mice were placed in a plexiglass beaker (20 cm in diameter and 40 cm high) containing water (25 °C) filled to a depth of 30 cm, in which they could not touch the bottom of the beaker. The swim test lasted a maximum of 6 min. Behavior was captured using a video camera, and experimenters monitored the animal's well-being throughout. Water was replaced and containers cleaned between individual mice. The durations of floating (immobility), climbing and swimming were scored manually for the final 4 min of recording. After testing, mice were towel dried and placed in a prewarmed cage before being returned to their home cage.

### Immunohistochemistry (IHC)

Perfused brains of CRH-ires-Cre mice (PN 90-120) were post-fixed in 4% paraformaldehyde in 0.1 M PBS (pH = 7.4) for 4–6 h, before cryoprotection in a 25% sucrose solution. Brains were frozen, then sectioned coronally into 20 μm thick slices using a Leica CM1900 cryostat (Leica Microsystems, Germany). To prepare tissue for the CRH staining, three CRH-Cre mice that had been injected with the Cre-driven AAV retrograde tracer in the NAc also received a single intracerebroventricular injection (AP: 0.49, ML: −0.70, DV: 2.70) of colchicine (10 μg/1 μl saline) (cat # 1364, Tocris Bioscience, UK) 48 h prior to perfusion to enhance CRH peptide localization in neuronal cell bodies. Coronal sections (20 μm) were subjected to CRH IHC using a tyramide signal amplification (TSA) technique[52]. Briefly, after several washes with 0.01 M PBS containing 0.3% Triton X-100 (PBS-T), sections were treated for 30 min in 0.3% $H_2O_2$-PBS-T, followed by blockade of nonspecific sites with 5% normal goat serum in PBS-T for 30 min. After rinsing with PBS three times, 5 min per wash, on a shaker at 95 rpm, sections were incubated for 48 h at 4 °C with rabbit anti-CRH primary antibody (courtesy of Paul E. Sawchenko, Salk Institute, PBL#rC68) in the blocking solution (dilution factor 1:20,000). The sections were rinsed with PBS-T three times, 5 min per wash, followed by incubation in a 0.5% blocking buffer (Cat # FP1020; PerkinElmer, US) for 1 h, and then in horseradish peroxidase (HRP)-conjugated anti-rabbit IgG (1:1000; Cat # NEF812, Lot # 050841, PerkinElmer, US) for 1.5 h. Fluorescein or cyanine 3-conjugated tyramide was diluted (1:150, Cat #'s NEL701A001KT and NEL704A001KT, respectively) in amplification buffer (Akoya Biosciences, US) and was applied in the dark for 5–6 min on ice. Sections were counter-stained with 10 μM DAPI (Cat # D-9542, Sigma-Aldrich, US), then wet-mounted on microscope slides and cover-slipped with

mounting medium (Vectashield, Cat # H-1200, Vector Labs, US). To assess the fidelity of CRH and the reporter, concurrent visualization of CRH and tdTomato was performed[52]. Sections (20 μm) were subjected to CRH IHC as described above. Fluorescein-conjugated tyramide (1:150, Cat # NEL701A001KT, Akoya Biosciences, US) was applied in the dark for 5–6 min on ice.

## In situ hybridization (ISH) and combined ISH and IHC

Perfused brains of CRH-ires-Cre mice (PN 90-120) were post-fixed in 4% paraformaldehyde in 0.1 M PBS (pH = 7.4) for 4–6 h, before cryoprotection in a 25% sucrose solution. Brains were frozen, then sectioned coronally into 20 μm thick slices using a Leica CM1900 cryostat (Leica Microsystems, Germany). The expression of Cre in the CRH-ires-CRE mouse line was detected via ISH, which was performed on free-floating brain sections[51]. Digoxigenin (DIG)–5′-conjugated Cre sense and anti-sense RNA oligonucleotide probes were generated (GenScript, US). The probe sequences used were: Antisense–5′-CCCUUCCAGGGCGC-GAGUUG-3′; Sense–5′-GGACACAGUGCCCGUGUCGG-3′. Sections fixed with 4% PFA were washed with 2× saline sodium citrate (SSC: 0.3 M NaCl, 0.03 M NaCitrate) and incubated in a solution composed of 2× SSC and prehybridization solution (1:1) at room temperature (RT) for 1 h prior to hybridization. The prehybridization solution consisted of 50% formamide, 4× SSC, 5× Denhardt's solution, 5% dextran sulfate, 100 μg/ml yeast tRNA and 100 μg/ml salmon sperm DNA. DIG-labeled probes were added into the prehybridization solution (1 μM) and hybridization was performed at 64 °C for 15 h in a humid chamber. After hybridization, sections were washed in 2× SSC at RT (2 × 15 min), 50% formamide/2× SSC at 70 °C for 1 h, 50% formamide/0.1× SSC at 70 °C for 1 h, and 0.1× and 0.05× SSC at 70 °C, each for 30 min. Hybrid molecules were detected with anti-DIG mouse antibody (1:2000, Cat # MAB7520, Lot # CGBN0218061, R&D Systems, US), followed by biotinylated anti-mouse IgG (1:400, Cat # BA-9200, Lot # ZE0924, Vector Labs, US) and avidin-biotin-peroxidase complex solution (1:200, Cat # PK-6100, Vector Labs, US)[52]. The sections were developed in 0.04% 3,3′-diaminobenzidine containing 0.01% $H_2O_2$ (DAB-Co Substrate Kit, Cat # 003843, Bioenno Tech, US). The specificity of the hybridization reaction was verified by substituting labeled sense probe for the antisense probe and by omitting the antisense probe, or performing on brain sections from C57Bl/6 mice. No labeling was observed under these conditions.

For dual labeling, brain sections were first processed for Cre ISH as described above. Hybrid molecules were detected with anti-DIG mouse antibody (1:2,000, Cat # MAB7520, Lot # CGBN0218061, R&D Systems, US), and cyanine 3-conjugated tyramide (1:150, Cat # NEL704A001KT, Akoya Biosciences) was applied. Sections were rinsed in PBS-T and then processed for CRH (1:20,000, PBL#rC68) or ChR2-EYFP (rabbit anti-GFP in 1:2000, Cat # 2555 S, Lot # 2, Cell Signaling, US) immunostaining. CRH signal was visualized with fluorescein-conjugated tyramide (1:150, Cat # NEL701A001KT, Akoya Biosciences) and ChR2-EYFP signal was visualized using anti-rabbit IgG conjugated to Alexa Fluor 488 (1:200, Cat # A-11034, Lot # 1971418, Invitrogen, US). Sections were mounted on gelatin-coated slides, air dried, and cover-slipped. To evaluate the possibility of altered sensitivity or specificity resulting from combined ISH and IHC, sections processed only for IHC or for ISH were compared with matched sections processed for dual labeling. No differences in labeled cells were observed.

## Image acquisition

Virus-labeled sections were scanned under a 10× objective of a fluorescent microscope (Nikon Eclipse E400) equipped with a high-sensitivity CCD camera (DS-Fi3). Confocal microscopy was utilized for imaging virus-labeled neurons after anterograde tracing (LSM 510 and 700, Zeiss, US). Images were obtained using ImageJ2 image acquisition software and Adobe Photoshop (CS5).

## Statistical analysis

All experiments and data analyses were conducted blind to experimental groups. The number of replicates (n) is indicated in figures as lines/dots and in figures legends and refers to the total number of experimental subjects independently treated in each experimental condition. Data are presented are mean values, and where applicable, accompanied by SEM. Statistical comparisons were performed using Prism 9 software (GraphPad, USA). Unpaired t-tests, paired t-tests, unpaired U-tests, and two-way ANOVA with repeated measures were used to test for statistical significance when appropriate. No statistical methods were used to pre-determine sample sizes. Statistical significance was set at $\alpha = 0.05$. P-values are provided in all figures and legends. Experiments were replicated in at least two independent batches, and from at least four different litters that yielded consistent results.

## Reporting summary

Further information on research design is available in the Nature Portfolio Reporting Summary linked to this article.

## Data availability

All the raw and/or processed data presented in this manuscript are available upon request. All mouse illustrations included in the main and supplementary figures were created with BioRender.com. Source data are provided with this paper.

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

## Acknowledgements

This work was supported by National Institute of Health grants P50 MH096889 (T.Z.B.), MH73136 (T.Z.B.), NS108296 (T.Z.B.), 1U01DA053826 (T.Z.B., S.V.M.), P50 DA044118 (S.V.M.), P50 MH096889 Seed Award FG23670 (L.Y.C.), The Bren Foundation (T.Z.B.), University of California-Irvine Startup Fund GF15247 (L.Y.C.), a George E. Hewitt Foundation for Biomedical Research Fellowship (M.T.B., T.Z.B.) and a British Society for Neuroendocrinology Project Support Grant BSN-5646342 (M.T.B.).

## Author contributions

M.T.B., A.K.S., B.G.G., L.Y.C., S.V.M., and T.Z.B. contributed to study design. M.T.B., A.K.S., G.B.d.C., L.T., and Y.C. contributed to data collection. M.T.B., L.T., and Y.C. conducted all surgeries. M.T.B., L.T., and A.K.S. conducted all behavior studies. Y.C., M.T.B., and A.L.P. conducted histological analyses. C.A.I. and X.X. provided anterograde and retrograde viruses. M.T.B. and T.Z.B. wrote the paper. All authors discussed and commented on the manuscript.

## Competing interests

The authors declare no competing interests.
