## [Peer Review File · Nature Communications]

Stress-induced plasticity of a CRH/GABA projection disrupts reward behaviors in mice.REVIEWER COMMENTS

Reviewer #1 (Remarks to the Author):

The manuscript purports to identify a novel population of CRH containing GABAergic neurons in the BLA that project to the Nucleus accumbens. Activation of these neurons appears to suppress reward in naive mice, and inhibition appears to reverse some effects of early life adversity. This is a very provocative study that details a previously un-identified population of neurons. There are a few issues that I think need to be dealt with to more convincingly show that this model holds water in the way that the authors conceptualize it.

1) The authors must do a dual in situ study exploring Cre and Crh so expression in the BLA. This is an exceedingly important validation step, and while the IHC shown is convincing, it does not sufficiently quantify this mouse line as a tool for exploring this pathway. Based on the image in pane 1g, there are many Crh containing neurons, or at least cre positive neurons, in the BLA. This seems much higher than anticipated.

2) The ephys presented in the Nac is compelling. it would be useful to provide some characterization of this novel population of neurons in the BLA as well, since they are not described to date.

3) It is unclear why the authors chose AAV1 for the optogenetic experiments, as this can have anterograde actions. Further, it would be of value to show maps of all injection sites for the viral experiments.

4) I am somewhat confused by Figure 4. The authors show robust effects of ELA on behavior in panels b,c,d, but these apparent differences are absent in the panels below. Further, it would seem that a 2 way anova would be the more appropriate analysis for examining DREADD impact on 2 treatment groups.

5) as a generic question, the authors don't really address the role of CRF vs GABA in the modulatory actions of this circuit, this seems like an important measure to study, especially given the idea stated in the introduction.

Reviewer #2 (Remarks to the Author):

The manuscript by Birnie et al., characterizes and elucidates the function of a novel GABAergic, CRH expressing BLA=>NAc pathway. They first characterize this pathway anatomically and functionally. They go on to show that either chemogenetic or optogenetic stimulation of this pathway in typically reared mice reduces reward-related behaviors, while chemogenetic inhibition has no significant effect. Conversely, in mice exposed to early life adversity (LBN), chemogenetic inhibition of the CRH^GABA BLA=>Ac pathway rescues the deleterious effects of ELA exposure. This indicates that there is a stress-induced plasticity that is occurring at this circuit. Importantly, these manipulations in the CRH^GABA

BLA=>NAc pathway specifically effect reward related behaviors and not other “unrewarded” behavior measures of anxiety-like behavior.

This is a very exciting and interesting study. It elucidates a GABAergic projection pathway from BLA to NAc that suppresses reward related behavior. This is very novel since, as the authors rightly point out, much of the scientific focus on the BLA=>NAc has been on the glutamatergic projection and promotes reward related behaviors. The trouble comes when the authors want to make assertions regarding CRH itself based on the present data. This is a typical and difficult conundrum since there are by no means trivial or perfect methods for testing whether the behavioral effects are actually due to CRH release from the BLA \ NAc. Given the experimenters’ technical repertoire demonstrated in the paper, the “simplest” way to test this would be to optogenetically stimulate CRH BLA neurons following microinfusion of CRH receptor (R1 or R1/R2 mixed) antagonist. I appreciate that this is an imperfect experiment since it requires stimulation in the BLA, leading to the release of transmitter in possibly more than just the NAc. This might be mitigated a bit if the experimenters use a retroAAV-ChR2 construct injected into the NAc of Crh-IRES-Cre mice, place the optogenetical fiber in the BLA and the cannula in the NAc. I appreciate that this is not an easy experiment to do, but I think is critical if the experimenters want to make claims about CRH itself.

Alternatively, the authors could greatly tone down the emphasis on CRH, beginning with switching the abbreviation to GABA^{CRH} BLA=>NAc. They could treat CRH as simply a genetic marker/tool with which to label this GABAergic projection circuit.

Can the authors conclude that CRH is only expressed in projection neurons and not GABAergic interneurons within the BLA? This would be an important point to make in terms of then justifying use of Crh-IRES-Cre line to isolate these GABAergic projection neurons (Option 2).

The other major issue with this study is that it was only performed in males. Clearly there are important sex differences in stress responsivity that need to be studied and highlighted. It must be the case that females are not always relegated to a “follow-up” study down the road or that happens never. It is not acceptable to continue to accept publications into high profile journals that only use one sex. Frankly in some ways, the study would have been stronger if they had used males and females, but only stuck to either chemogenetics or optogenetics.

While I perceive these two points as the major issues with this study, there are other, more moderate concerns that also need to be addressed prior to publication, which include additional experiments.

1. Demonstration of stress-induced plasticity at CRHGABA BLA=>NAc synapse: These authors show nice electrophysiological validation of exclusively GABAergic nature of the projection. The behavior shown in Figure 4 indicates that there has been an increase in synaptic strength of this pathway that is being reversed by Gi-DREADD activation. It would greatly strengthen the manuscript and support the conclusions if they actually demonstrated that there was a LTPGABA phenomenon or even change in PPR properties/quantal release properties occurring at this synapse in ELA exposed mice compared to TR mice.

2. Gq-DREADD in ELA exposed mice: Not sure if somehow, I'm missing it (apologies if I am), but I'm not seeing the Gq-DREADD manipulation in ELA mice. It is important and interesting that in TR mice, Gi-DREADD is not effective. This gives us important information regarding the dynamics of this input. Likewise, it would be important to know whether Gq-DREADD activation in ELA mice would exacerbate the impact of ELA. This should be done.

3. Quantification of enhanced cfos following Gq-DREADD. It would be appropriate to perform quantification of this enhanced cfos response to go along with the image. The validation would be strengthened if the complimentary study with Gi-DREADD were also included.

Minor

4. I would suggest Figure 4g,h be moved up to Figure 3. I believe, given the statistics, this may be a better place for it. Otherwise, if it is kept in Figure 4, it seems as though it would need to be directly compared to ELA GiDREADD experiments via a 2 way ANOVA.

5. It would be useful to move Supp Figure 1a,b to Figure 3.

6. The comparison made in Supp Figure 2f is not appropriate. I would remove that graph.

7. Line 26-30 of the abstract: Very long, confusing run-on sentence. Please consider re-wording.

Reviewer #3 (Remarks to the Author):

This manuscript by Birnie et al describes an interesting study of the role of a novel BLA-NAcc projection in mimicking and correcting reward behaviors impaired by early life deprivation experiences.

The authors identify this novel BLA-NAc projection and show it co-releases CRH and GABA to produce inhibitory effects. They go on to show that both chemogenetic and optogenetic stimulation of this projection in typically reared control mice replicates inhibitory effects of early life deprivation on reward related behaviors. They further show that DREADD inhibition of the projection in early deprived mice restored normal consumption of palatable food and restored the preference for a sex-related cue, making these reward behaviors more similar to control mice.

As the authors conclude, this insight into BLA-NAc mechanisms of stress-mediated influences of reward behavior may be relevant to understanding the mechanisms of stress-related mental illnesses in humans.

Altogether, the experiments are well designed, and the results seem compelling. I believe this is a powerful and interesting set of results that is likely to be of wide interest.

Suggestion:

While stress and CRH mechanisms do often inhibit reward motivation, as the authors show here there have also been reports that CRH activation in NAc can oppositely facilitate reward related behaviors, at

least under some conditions (e.g., Lemos et al. *Neuropsychopharmacology* 2020; Lemos et al. *Nature* 2012; Pecina et al. *Current Biology* 2006). Although the authors' BLA-NAc CRH effect here is clearly different, it would be good to at least acknowledge that opposite role. Perhaps a brief discussion of possible factors that might determine whether NAc CRH exerts pro-reward vs anti-reward effects (e.g., co-release of GABA with CRH; testing conditions; etc?) would be helpful for placing these interesting new results in broader context and strengthen the impact of the manuscript.

Response to Reviewers:

Reviewer #1 (Remarks to the Author):

The manuscript purports to identify a novel population of CRH containing GABAergic neurons in the BLA that project to the Nucleus accumbens. Activation of these neurons appears to suppress reward in naive mice, and inhibition appears to reverse some effects of early life adversity. This is a very provocative study that details a previously un-identified population of neurons. There are a few issues that I think need to be dealt with to more convincingly show that this model holds water in the way that the authors conceptualize it.

We appreciate the Reviewer's enthusiasm for the work here, and for their important points to enhance the manuscript, which we have addressed below:

1) The authors must do a dual in situ study exploring Cre and Crh so expression in the BLA. This is an exceedingly important validation step, and while the IHC shown is convincing, it does not sufficiently quantify this mouse line as a tool for exploring this pathway. Based on the image in pane 1g, there are many Crh containing neurons, or at least cre positive neurons, in the BLA. This seems much higher than anticipated.

We thank the Reviewer for this suggestion to enhance the rigor of our work. Accordingly, as shown in the new Supplemental Figure 1, we performed dual in situ hybridization / immunocytochemistry studies for Cre and CRH, respectively, and colocalized Cre to neurons expressing the native CRH expressing neurons in the BLA of CRH-ires-Cre mice (and very few others). In addition, we now demonstrate also a selective Cre localization with CRH-ires Cre-driven channelrhodopsin expression used for projection manipulation in this study. These new data (as well as the testing of the Cre construct) are found in Suppl. Figure 1.

2) The ephys presented in the Nac is compelling. it would be useful to provide some characterization of this novel population of neurons in the BLA as well, since they are not described to date.

We concur with the Reviewer that electrophysiological data exploring the activity, characteristics, and inputs to the CRH⁺ BLA neurons is an exciting and important avenue of research. The comprehensive characterization of the BLA CRH neurons (including those that project to the NAc) projection is a subject of an ongoing large and distinct study over the coming year.

3) It is unclear why the authors chose AAV1 for the optogenetic experiments, as this can have anterograde actions.

We regret the apparent ambiguity and resulting confusion. We concur with the Reviewer that AAV1 is optimal for anterograde studies. Accordingly, we injected AAV1 virus into the BLA to anterogradely label projections of BLA CRH⁺ cells. We then stimulated the BLA-NAc projection selectively by the use of optic fibers in the NAc.

Further, it would be of value to show maps of all injection sites for the viral experiments.

We concur, with the Reviewer, and now include maps of the injection sites as well as the optic fiber/cannula terminal locations in Suppl. Figure. 4.

4) I am somewhat confused by Figure 4. The authors show robust effects of ELA on behavior in panels b,c,d, but these apparent differences are absent in the panels below.

We appreciate the Reviewer's concern and concur that there is some within-experiment variability in the absolute amount of palatable food consumed, though there is always a suppression of reward behaviors in ELA mice vs controls. More specifically, a difference in Figure 4, panel b vs panel e mice is that the ELA mice in b did not receive local injections into the NAc, which might influence consumption. More likely is a bit of batch variation (these are not litter effects; we ascertain that each experiment includes mice from several litters). Indeed, the new experiments performed in response to Reviewers' suggestions again show a robust suppression of palatable food consumption, even upon introduction of CNO to the NAc: please see Figure 4g, h - hM3Dq DREADD

Further, it would seem that a 2 way anova would be the more appropriate analysis for examining DREADD impact on 2 treatment groups.

We thank the Reviewer for this suggestion. Accordingly, and based also on input from Reviewer 2, we have reworked the figures, substantially influencing the analytic methodologies.

5) as a generic question, the authors don't really address the role of CRF vs GABA in the modulatory actions of this circuit, this seems like an important measure to study, especially given the idea stated in the introduction.

We concur with the Reviewer that uncovering the relative roles of CRF and GABA, both released by our novel projection, is of significant interest. This major undertaking is indeed a direction of ongoing and future studies.

Reviewer #2 (Remarks to the Author):

The manuscript by Birnie et al., characterizes and elucidates the function of a novel GABAergic, CRH expressing BLA=>NAc pathway. They first characterize this pathway anatomically and functionally. They go on to show that either chemogenetic or optogenetic stimulation of this pathway in typically reared mice reduces reward-related behaviors, while chemogenetic inhibition has no significant effect. Conversely, in mice exposed to early life adversity (LBN), chemogenetic inhibition of the CRH^GABA BLA=>NAc pathway rescues the deleterious effects of ELA exposure. This indicates that there is a stress-induced plasticity that is occurring at this circuit. Importantly, these manipulations in the CRH^GABA BLA=>NAc pathway specifically effect reward related behaviors and not other "unrewarded" behavior measures of anxiety-like behavior.

This is a very exciting and interesting study. It elucidates a GABAergic projection pathway from BLA to NAc that suppresses reward related behavior. This is very novel since, as the authors rightly point out, much of the scientific focus on the BLA=>NAc has been on the glutamatergic projection and promotes reward related behaviors.

We thank the Reviewer for the enthusiasm for the novelty and impact of this study. We also appreciate the comments and suggestions, and have addressed each one below.

The trouble comes when the authors want to make assertions regarding CRH itself based on the present data. This is a typical and difficult conundrum since there are by no means trivial or perfect methods for testing whether the behavioral effects are actually due to CRH release from the BLA-NAc. Given the experimenters' technical repertoire demonstrated in the paper, the "simplest" way to test this would be to optogenetically stimulate CRH BLA neurons following microinfusion of CRH receptor (R1 or R1/R2 mixed) antagonist. I appreciate that this is an imperfect experiment since it requires stimulation in the BLA, leading to the release of transmitter in possibly more than just the NAc. This might be mitigated a bit if the experimenters use a retroAAV-ChR2 construct injected into the NAc of Crh-IRES-Cre mice, place the optogenetical fiber in the BLA and the cannula in the NAc. I appreciate that this is not an easy experiment to do, but I think is critical if the experimenters want to make claims about CRH itself.

We concur with the Reviewer (as well as with Reviewer 1) that teasing out the relative contribution of CRH and GABA to the function of the projection is complex. Therefore, we do not ascribe in the current paper specific roles for one of these two mediators. Indeed, they may actually work in tandem or synergistically (e.g., CRH influences GABA release; Herman et al., 2016 – PMID 27798128). Instead, we describe the sum effects of the novel projection as a whole.

As mentioned also in the response to Reviewer 1, resolving CRH- from GABA effects is an important and highly complex set of experiments which we hope to implement. The Reviewer correctly indicates potential caveats in several approaches. Specifically, targeting CRH in the amygdala might not be optimal, as we now find that BLA-CRH neurons also project to other brain regions, so that optogenetically stimulating the BLA would have potential off-target effects

Alternatively, the authors could greatly tone down the emphasis on CRH, beginning with switching the abbreviation to GABA^{CRH} BLA=>NAc. They could treat CRH as simply a genetic marker/tool with which to label this GABAergic projection circuit. Can the authors conclude that CRH is only expressed in projection neurons and not GABAergic interneurons within the BLA? This would be an important point to make in terms of then justifying use of Crh-IRES-Cre line to isolate these GABAergic projection neurons (Option 2).

We appreciate the Reviewer's viewpoint regarding the potential limited role of CRH in the novel projection. In accord with providing the best representation of available information, we indeed changed our nomenclature for the projection to: CRH/GABA BLA-NAc projection. In addition, we toned down our premise and rationale for investigating CRH⁺ projections as, suggested by the Reviewer:

"Therefore, considering the peptide a potential marker or an effector in stress-sensitive neurons, we mapped CRH- expressing afferent projections to the NAc" (line 58).

The Reviewer's suggestion that the peptide may simply be a genetic marker for this GABAergic projection, however, does not seem to be supported by the facts. There are other populations of CRH/GABA or CRH/PV expressing neurons in the BLA. Some may be local interneurons, others seem to project to the BNST (Interestingly, these are not located in the medial BLA, a region shown by Beyeler and colleagues to project to NAc (PMID: 29386133). Instead, they are in the dorsal lateral BLA). Thus, we join the Reviewer in concluding that the specific and relative roles of GABA

and CRH in the function of the CRH/GABA projection are not yet known and will be studied imminently. Importantly, in the current work, we have clearly established that (1) projection is distinct in its origin and termination from other BLA/GABA/CRH neurons and (2) its function is dependent on local release of its modulators within the NAc, the location of our interventions.

The other major issue with this study is that it was only performed in males. Clearly there are important sex differences in stress responsivity that need to be studied and highlighted. It must be the case that females are not always relegated to a “follow-up” study down the road or that happens never. It is not acceptable to continue to accept publications into high profile journals that only use one sex. Frankly in some ways, the study would have been stronger if they had used males and females, but only stuck to either chemogenetics or optogenetics.

We thank the Reviewer for this query, and strongly endorse the need to examine both sexes (a view shared by the Editor). Accordingly, we had carried out experiments in both males and females. In females, we discovered that stimulating the projection in typically-reared animals had little effect, showing the remarkable complexity of sex-dependent mechanisms of reward behaviors and the effects of early-life adversity.

To directly address the Reviewer’s query, we have extended these experiments and now provide three new sets of data:

- a. Figure 3d now includes females, showing that stimulation of the projection does not influence palatable food consumption.
- b. Figure 3e now includes females showing that stimulation of the projection does not influence sex-cue preference.
- c. In analogy to the data in males, suppl. Figure 2b now shows stimulation of the CRH⁺ BLA cells also does not affect reward behavior.

In the aggregate, these data suggest that the influence of early-life adversity on female reward behaviors (which is behaviorally distinct from the effects on males, e.g., Levis et al., Mol Psych, 2021 – PMID: 31822817) may be mediated via distinct components of the reward circuitry and are not directly influenced by the function of the CRH/GABA BLA-NAc projection. We have added Ms. Lara Taniguchi, who is pursuing the female studies, to the list of authors.

While I perceive these two points as the major issues with this study, there are other, more moderate concerns that also need to be addressed prior to publication, which include additional experiments.

1. Demonstration of stress-induced plasticity at CRHGABA BLA=>NAc synapse: These authors show nice electrophysiological validation of exclusively GABAergic nature of the projection. The behavior shown in Figure 4 indicates that there has been an increase in synaptic strength of this pathway that is being reversed by Gi-DREADD activation. It would greatly strengthen the manuscript and support the conclusions if they actually demonstrated that there was a LTPGABA phenomenon or even change in PPR properties/quantal release properties occurring at this synapse in ELA exposed mice compared to TR mice.

We appreciate the Reviewer’s suggestion for these exciting future experiments. We share the Reviewer’s assumption that the behavioral phenomena may indeed be a result of synaptic plasticity. However, these outcomes may also result from many other mechanisms, including an increase in

the number of CRHR1 receptors, a switch of receptor role, and a variety of GABAergic changes. Therefore, the proposed experiments, which will require many months, might not yield synaptic plasticity as the ultimate mechanism for the striking behavioral effects of the projection after ELA.

2. Gq-DREADD in ELA exposed mice: Not sure if somehow, I'm missing it (apologies if I am), but I'm not seeing the Gq-DREADD manipulation in ELA mice. It is important and interesting that in TR mice, Gi-DREADD is not effective. This gives us important information regarding the dynamics of this input. Likewise, it would be important to know whether Gq-DREADD activation in ELA mice would exacerbate the impact of ELA. This should be done.

The Reviewer's point is well taken. It was our oversight in not performing these experiments in the original submission. In accord with the Reviewer's recommendation, we now include the results of these new experiments in the revised Figure 4, panels g and h.

3. Quantification of enhanced cfos following Gq-DREADD. It would be appropriate to perform quantification of this enhanced cfos response to go along with the image. The validation would be strengthened if the complimentary study with Gi-DREADD were also included.

We appreciate the Reviewer's point, and we now provide the quantification of Fos expression as a measure of neuronal activation by CNO in neurons expressing the excitatory DREADD, hM3Dq (Supplemental Figure 2C, both panels).

Minor

4. I would suggest Figure 4g,h be moved up to Figure 3. I believe, given the statistics, this may be a better place for it. Otherwise, if it is kept in Figure 4, it seems as though it would need to be directly compared to ELA GiDREADD experiments via a 2 way ANOVA.

As suggested by the Reviewer, we have moved the typical rearing (TR) GiDREADD data to Figure 3f, g

5. It would be useful to move Supp Figure 1a,b to Figure 3.

As recommended, we now show the brain coordinates (original suppl. Figure. 1a,b) in Figure 3a, b.

6. The comparison made in Supp Figure 2f is not appropriate. I would remove that graph.

We have removed this graph.

7. Line 26-30 of the abstract: Very long, confusing run-on sentence. Please consider re-wording.

As suggested, we modified the text and hope it is clearer.

Reviewer #3 (Remarks to the Author):

This manuscript by Birnie et al describes an interesting study of the role of a novel BLA-NAcc projection in mimicking and correcting reward behaviors impaired by early life deprivation experiences.

The authors identify this novel BLA-NAc projection and show it co-releases CRH and GABA to produce inhibitory effects. They go on to show that both chemogenetic and optogenetic stimulation of this projection in typically reared control mice replicates inhibitory effects of early life deprivation on reward related behaviors. They further show that DREADD inhibition of the projection in early deprived mice restored normal consumption of palatable food and restored the preference for a sex-related cue, making these reward behaviors more similar to control mice.

As the authors conclude, this insight into BLA-NAc mechanisms of stress-mediated influences of reward behavior may be relevant to understanding the mechanisms of stress-related mental illnesses in humans.

Altogether, the experiments are well designed, and the results seem compelling. I believe this is a powerful and interesting set of results that is likely to be of wide interest.

We deeply thank the Reviewer for the appreciation and strong enthusiasm for our studies.

Suggestion:

While stress and CRH mechanisms do often inhibit reward motivation, as the authors show here there have also been reports that CRH activation in NAc can oppositely facilitate reward related behaviors, at least under some conditions (e.g., Lemos et al. Neuropsychopharmacology 2020; Lemos et al. Nature 2012; Pecina et al. Current Biology 2006). Although the authors' BLA-NAc CRH effect here is clearly different, it would be good to at least acknowledge that opposite role. Perhaps a brief discussion of possible factors that might determine whether NAc CRH exerts pro-reward vs anti-reward effects (e.g., co-release of GABA with CRH; testing conditions; etc?) would be helpful for placing these interesting new results in broader context and strengthen the impact of the manuscript.

We concur heartily with the Reviewer. The actions of CRH are intricate and context dependent, leading to even opposite behaviors depending on different inputs and/or possible stress challenges. We have embraced this notion and now include it in the Discussion section.

REVIEWERS' COMMENTS

Reviewer #1 (Remarks to the Author):

The authors have done mostly a great job in replying to my comments. However, one concern remains. There is still no quantification of cre/crh. The added image is nice (Figure S1), but this should be quantified.

Reviewer #2 (Remarks to the Author):

Overall, the authors have nicely addressed my concerns with discussion and additional experiments. Having read the responses to the other reviewers, I would recommend the following prior to the final publication. However, these are suggestions and not required for me to accept the final version of the paper.

1. In Response to Reviewer 1, the authors should provide quantification of the co-expression of the Cre mRNAs with CRF-ir, not just an example image.

2. Regarding Reviewer 1's request for an electrophysiological characterization of BLA CRF+ neurons, I agree with the authors that a full characterization would be the subject of subsequent studies. However, I think that understanding basic features of cell excitability such as the rheobase and spontaneous firing of these neurons would be really important to know since it will affect the stimulation necessary for causing the release of CRF as a neuropeptide neurotransmitter.

Reviewer #3 (Remarks to the Author):

In this revision and cover letter, the authors have been very responsive to comments raised in the initial reviews. Their revisions have strengthened an already strong manuscript and interesting study. I have no further objections.

REVIEWERS' COMMENTS

We would like to thank all the Reviewers for their important suggestions which have strengthened this manuscript.

Reviewer #1 (Remarks to the Author):

The authors have done mostly a great job in replying to my comments. However, one concern remains. There is still no quantification of cre/crh. The added image is nice (Figure S1), but this should be quantified.

We agree with the Reviewer and have now included the quantification, located in Supplementary Figure 1, with the cre/crh images.

Reviewer #2 (Remarks to the Author):

Overall, the authors have nicely addressed my concerns with discussion and additional experiments. Having read the responses to the other reviewers, I would recommend the following prior to the final publication. However, these are suggestions and not required for me to accept the final version of the paper.

1. In Response to Reviewer 1, the authors should provide quantification of the co-expression of the Cre mRNAs with CRF-ir, not just an example image.

We thank the Reviewer for this. In response to the Reviewers, we have quantified the cre/crh colocalization and have included this within Supplementary Figure 1.

2. Regarding Reviewer 1's request for an electrophysiological characterization of BLA CRF+ neurons, I agree with the authors that a full characterization would be the subject of subsequent studies. However, I think that understanding basic features of cell excitability such as the rheobase and spontaneous firing of these neurons would be really important to know since it will affect the stimulation necessary for causing the release of CRF as a neuropeptide neurotransmitter.

This is an excellent point. We concur that understanding the features of the CRH/GABA+ BLA cells is important., and is part of a comprehensive electrophysiological study of the CRH+ BLA-NAc projection that we are currently involved in.

Reviewer #3 (Remarks to the Author):

In this revision and cover letter, the authors have been very responsive to comments raised in the initial reviews. Their revisions have strengthened an already strong manuscript and interesting study. I have no further objections.

We thank the Reviewer for their support of this study.